# LEARNING SEQUENCE ATTRACTORS IN HOPFIELD NETWORKS WITH HIDDEN NEURONS

## ABSTRACT

The brain is targeted for processing temporal sequence information. It remains largely unclear how the brain learns to store and retrieve sequence memories. Here, we study how networks of Hopfield type learn sequence attractors to store predefined pattern sequences and retrieve them robustly. We show that to store arbitrary pattern sequences, it is necessary for the network to include hidden neurons even though their role in displaying sequence memories is indirect. We develop a local learning algorithm to learn sequence attractors in the networks with hidden neurons. The algorithm is proven to converge and lead to sequence attractors. We demonstrate that our model can store and retrieve sequences robustly on synthetic and real-world datasets. We hope that this study provides new insights in understanding sequence memory and temporal information processing in the brain.

## 1 INTRODUCTION

The brain is targeted for processing temporal sequence information. Taking visual recognition for example, the conventional setting of static image processing never happens in the brain. Starting from retina, visual inputs of an image arrive in the form of optical flow, which are transformed into spike trains of retinal ganglia cells, and then transmitted through LGN, V1, V2, V4 and higher cortical regions until the image is recognized. Along the whole pathway, the computations performed by the brain are in the form of temporal sequential processing, rather than being static. For another example, when we recall an episodic memory, a sequence of events represented by neuronal responses flows into our mind, and these events do not come in disorder or isolation, but are unfolded in time, as we experience "mental time travel" [Tulving, 2002]. The hippocampus has been revealed to be essential for sequence memories by physiological and behavioral studies. In animals, sequences of neural activity patterns are observed in the hippocampus for memory replay and memory related tasks [Nádasdy et al., 1999; Lee & Wilson, 2002; Foster & Wilson, 2006; Pastalkova et al., 2008; Davidson et al., 2009; Pfeiffer & Foster, 2013; 2015]. The discovery of time cells in hippocampus shows that the brain has specialized neurons encoding the temporal structure of events [Eichenbaum, 2014]. Overall, the processing of temporal sequences is critical to the brain, but computational modeling study on this issue lags far behind that on static information processing.

Attractor neural networks are promising computational models for elucidating the mechanisms of the brain representing, memorizing and processing information [Amari, 1972; Hopfield, 1982; Amit, 1989]. An attractor network is a type of recurrent networks, in which information is stored as stable states (attractors) of the network. Once stored, a memory can be retrieved robustly under the evolving of the network dynamics given noisy or incomplete cues. The experimental evidences have indicated that the brain employs attractor networks for memory related tasks [Khona & Fiete, 2022]. By considering simplified neuron model and threshold dynamics, the classical Hopfield networks have successfully elucidated how recurrent networks learn to store static memory patterns [Hopfield, 1982]. However, the classical Hopfield networks and some related works mostly consider static attractors, which can not explain the sequential neural activities widely observed in the brain (e.g., in memory retrieval in the hippocampus). Thus, in this paper, we follow and extend the standard form of the classical Hopfield networks by considering binary neurons and threshold dynamics, as this enables us to pursue theoretical analysis, and we investigate how the networks learn to store sequence attractors. By sequence attractor, it means the state of a recurrent network evolves in the order of the stored pattern sequence and being robust to noise.

We first show that, to store arbitrary pattern sequences, the classical Hopfield networks, which have only visible neurons, is inadequate in Section 3. Then we argue that it is necessary for the networks to include hidden neurons in Section 4. These neurons are not directly involved in expressing pattern sequences, but they are indispensable for the networks to store and retrieve arbitrary pattern sequences. When unfolding the network recurrent dynamics in time, the role of these hidden neurons is analogous to that of a hidden layer in the conventional feedforward networks. We further develop a local learning algorithm to learn sequence attractors in the networks with hidden neurons in Section 5. The algorithm is proven to converge and lead to sequence attractors. It draws inspirations from three ideas: feedback alignment [Lillicrap et al., 2016], target propagation [LeCun, 1987; Bengio, 2014; Litwin-Kumar et al., 2017] and three-factor rules [Frémaux & Gerstner, 2016; Kuśmierz et al., 2017]. We demonstrate that our model can learn to store and retrieve pattern sequences robustly on synthetic and real-world datasets in Section 6. We hope that this study provides new insights in understanding sequence memory and temporal information processing in the brain.

## 2 RELATED WORKS AND OUR CONTRIBUTIONS

Learning temporal sequences in recurrent networks has been studied previously in the field of computational neuroscience. These works employ different forms of recurrent networks and have different focuses of investigation. Specifically, [Amari, 1972; Hopfield, 1982; Kleinfeld, 1986; Sompolinsky & Kanter, 1986; Bressloff & Taylor, 1992; Fiete et al., 2010] investigated recurrent networks of binary neurons and simple threshold dynamics. This approach takes advantages of simplified models that capture the essential features of neural dynamics and allows us to pursue theoretical analysis. [Brea et al., 2013; Tully et al., 2016] investigated recurrent networks of spiking neurons which are more biologically realistic but hard to analyze theoretically. [Laje & Buonomano, 2013; Rajan et al., 2016; Gillett et al., 2020; Rajakumar et al., 2021] investigated recurrent networks of firing-rate neurons (e.g., sigmoid and linear-threshold), whose complexity is a trade-off between binary neurons and spiking neurons. More recently, [Karuvally et al., 2023; Chaudhry et al., 2023] employ modern Hopfield networks [Krotov & Hopfield, 2016] for modeling sequence memory.

In this paper, we follow and extend networks of Hopfield type, which uses binary neuron model, with the focus of theoretical analysis. Below summarizes the main contributions of our work in comparison to related works.

- We highlight the importance of hidden neurons in the networks for learning arbitrary pattern sequences. Most previous works on Hopfield networks of binary neurons considered only visible neurons, and hence the pattern sequences they can generate are limited [Amari, 1972; Kleinfeld, 1986; Sompolinsky & Kanter, 1986; Bressloff & Taylor, 1992; Fiete et al., 2010]. Although modern Hopfield networks also include hidden neurons, the biological plausibility is unclear as it requires a high-order polynomial or exponential activation function [Chaudhry et al., 2023]. Other works using biologically more detailed models did demonstrate the benefit of including hidden neurons in sequence learning, but they lack theoretical analysis because of the model complexity [Laje & Buonomano, 2013; Brea et al., 2013; Rajakumar et al., 2021].

- We have clear theoretical characterization of sequences that can be generated by our networks (Proposition 1), a result which is lacking in all other related works. Although this conclusion comes from the analysis of the simple model we use, it lays foundation for future work to test it in biologically more realistic networks.

- Our learning algorithm is proven to converge and lead to sequence attractors, while most previous works only provided empirical evidences to demonstrate that the learned sequences are robust to noise [Laje & Buonomano, 2013; Brea et al., 2013; Rajan et al., 2016; Rajakumar et al., 2021]. Although [Amari, 1972; Bressloff & Taylor, 1992] provided provable results on sequence attractors, they did not include hidden neurons and hence the results only hold for a restricted class of sequences.

- Our learning algorithm only requires local information between neurons, which is believed to be biologically plausible. [Rajakumar et al., 2021] used backpropagation which is often criticized for its biologically implausibility as it requires gradient computation and has the weight transport problem [Lillicrap et al., 2020]. [Chaudhry et al., 2023] used a pseudo-inverse algorithm whose biological plausibility is unclear.

## 3   LIMITATION OF CLASSICAL HOPFIELD NETWORKS

We first consider the classical Hopfield networks of $N$ visible binary neurons [Amari, 1972; Hopfield, 1982]. All the neurons are bidirectionally connected and their weight matrix is $\mathbf{W}$ of which $W_{ij}$ denotes the synaptic weight from the $j$-th neuron to the $i$-th neuron. Let $\boldsymbol{\xi}(t) = (\xi_1(t), ..., \xi_N(t)) \in \{-1, 1\}^N$ be the states of the neurons at time $t$. These states are synchronously updated according to the threshold dynamics, for $i = 1, ..., N$,

$$\xi_i(t+1) = \text{sign}\Big(\sum_{j=1}^{N} W_{ij}\xi_j(t) - \theta_i\Big) = \text{sign}\Big(\sum_{j=0}^{N} W_{ij}\xi_j(t)\Big), \tag{1}$$

where $\text{sign}(x) = 1$ if $x \geq 0$ and $\text{sign}(x) = 0$ otherwise. Hereafter, the threshold parameter $\theta_i$ is omitted as it can be absorbed by adding an extra neuron $\xi_0(t) = 1$ and $W_{i0} = -\theta_i$. Given a pair of successive network states $\boldsymbol{\xi}(t)$ and $\boldsymbol{\xi}(t+1)$, the dynamics of the network can be unfolded in time and viewed as a feedforward network, in which each output neuron is a perceptron of the inputs, as illustrated in Figure 1.

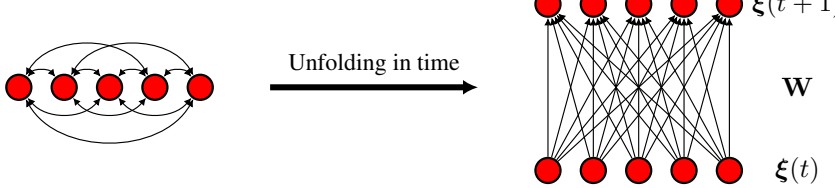

Figure 1: Classical Hopfield network. The red circles denote the visible neurons.

Given a sequence in the form of $\mathbf{x}(1), ..., \mathbf{x}(T) \in \{-1, 1\}^N$, one can use a learning algorithm to adjust $\mathbf{W}$ such that the evolving of the network state matches the pattern sequence. Although the networks can generate some sequences of maximal length $2^N$ [Muscinelli et al., 2017], they are fundamentally limited in the class of sequences that can be generated. Since each neuron can be regarded as a perceptron, the condition that sequence $\mathbf{x}(1), ..., \mathbf{x}(T)$ can be generated by the network is, for each $i$, the dataset $\{(\mathbf{x}(t), x_i(t+1))\}_{t=1}^{T-1}$ is linearly separable [LeCun, 1986; Bressloff & Taylor, 1992; Brea et al., 2013; Muscinelli et al., 2017].

For a simple example of sequences which cannot be generated by the networks, consider the sequence $(1, 1), (1, -1), (-1, 1), (-1, -1), (1, 1)$ with $N = 2$ and $T = 5$. To generate this sequence, the first neuron of the network needs to map $(1, 1)$ to $1, (1, -1)$ to $-1, (-1, 1)$ to $-1$ and $(-1, -1)$ to $1$. This mapping is essentially the XOR operation which cannot be performed by a perceptron [Minsky & Papert, 1969]. In Figure 2, we show additional examples of sequences that cannot be generated by the network. The sequences are synthetically constructed. We then test if the perceptron learning algorithm can learn the sequences. Since the algorithm converges if the linear separability condition is met [Minsky & Papert, 1969], the divergence of the algorithm implies that the sequences cannot be generated by the networks.

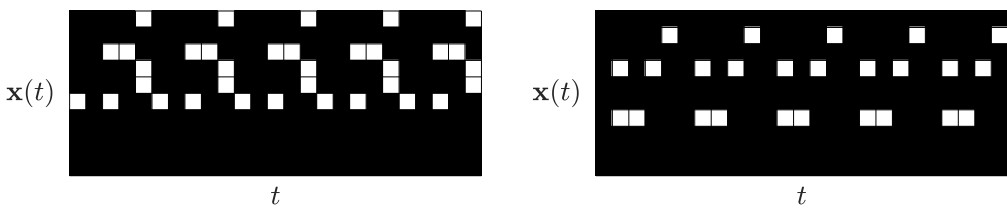

Figure 2: Two example sequences which cannot be generated by recurrent networks without hidden neurons. White squares denote positive ones and black squares denote negative ones.

## 4 Hopfield Networks with Hidden Neurons

To overcome the limitation of the classical Hopfield networks, we consider a group of hidden neurons in the network, in addition to visible ones. The visible and hidden neurons are bidirectionally connected, and there is no intra-connection within visible neurons or hidden neurons. Let $\mathbf{U}$ be the weight matrix from visible neurons to hidden neurons, of which $U_{ij}$ denotes the synaptic weight from the $j$-th visible neuron to the $i$-th hidden neuron, and $\mathbf{V}$ be the weight matrix from hidden neurons to visible neurons, of which $V_{ji}$ denotes the synaptic weight from the $i$-th hidden neuron to the $j$-th visible neuron. Let $\boldsymbol{\xi}(t) = (\xi_1(t), ..., \xi_N(t)) \in \{-1, 1\}^N$ be the states of visible neurons and $\boldsymbol{\zeta}(t) = (\zeta_1(t), ..., \zeta_M(t)) \in \{-1, 1\}^M$ be the states of hidden neurons at time $t$. These states are synchronously updated according to, for $i = 1, ..., M$ and $j = 1, ..., N$,

$$\zeta_i(t) = \text{sign}\Big( \sum_{k=1}^{N} U_{ik} \xi_k(t) \Big), \tag{2}$$

$$\xi_j(t+1) = \text{sign}\Big( \sum_{k=1}^{M} V_{jk} \zeta_k(t) \Big), \tag{3}$$

where we omit the threshold parameters as they can be absorbed into the equations. As illustrated in Figure 3, given a pair of successive network states $\boldsymbol{\xi}(t)$ and $\boldsymbol{\xi}(t+1)$, the dynamics of the network can be unfolded in time and viewed as a feedforward network with a hidden layer of neurons.

The networks of $M$ hidden neurons can generate arbitrary sequences with Markov property and of length at least $M$, as stated in Proposition 1. We provide a constructive proof based on one-hot encoding by the hidden neurons in the Appendix.

**Proposition 1** *Let $\mathbf{x}(1), ..., \mathbf{x}(T) \in \{-1, 1\}^N$ such that $\mathbf{x}(i) \neq \mathbf{x}(j)$ for $i \neq j$ except that $\mathbf{x}(1) = \mathbf{x}(T)$. Then $\mathbf{x}(1), ..., \mathbf{x}(T)$ can be generated by the network defined in (2)(3) for $M = T - 1$.*

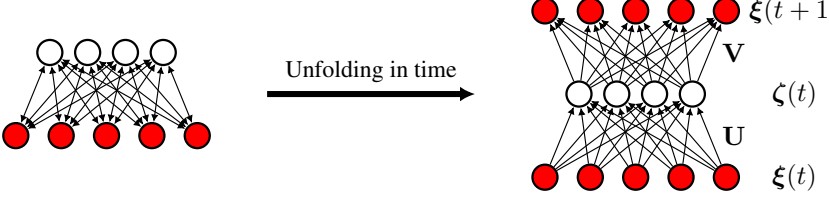

Figure 3: Hopfield network with hidden neurons. The red circles denote visible neurons and the white circles denote hidden neurons.

## 5 Learning

To learn the weight matrices, one can first unfold the Hopfield network with hidden neurons in time such that it becomes a feedforward network with a hidden layer, and the pairs of successive patterns in the sequence constitute the training examples. However, learning in the unfolded feedforward network is difficult since the backpropagation algorithm cannot be applied as the neurons are not differentiable.

We propose a new learning algorithm to learn the weight matrices in the unfolded feedforward networks, which draws inspirations from three ideas: feedback alignment [Lillicrap et al., 2016], target propagation [LeCun, 1987; Bengio, 2014; Litwin-Kumar et al., 2017] and three-factor rules [Frémaux & Gerstner, 2016; Kuśmierz et al., 2017]. As in feedback alignment, it requires a random matrix $\mathbf{P}$, which is fixed during the learning process, to backpropagate signals. As in target propagation, it does not propagate errors but targets to create surrogate targets for the hidden neurons. Each weight parameter is updated by a three-factor rule, in which the presynaptic activation, the postsynaptic activation and an error term as neuromodulation are multiplied. The three-factor rule is similar to the one for Hopfield networks without hidden neurons [Bressloff & Taylor, 1992] and known as margin perceptron in the machine learning literature [Collobert & Bengio, 2004].

The algorithm works as follows. Given a pair of successive patterns $\mathbf{x}(t)$ and $\mathbf{x}(t+1)$, for $i = 1, ..., M$ and $j = 1, ..., N$ in parallel,

1. Update $\mathbf{U}$ by

$$z_i(t+1) = \text{sign}\Big(\sum_{k=1}^{N} P_{ik} x_k(t+1)\Big), \tag{4}$$

$$\mu_i(t) = H\Big(\kappa - z_i(t+1)\sum_{k=1}^{N} U_{ik} x_k(t)\Big), \tag{5}$$

$$U_{ij} \leftarrow U_{ij} + \eta\mu_i(t)z_i(t+1)x_j(t). \tag{6}$$

2. Update $\mathbf{V}$ by

$$y_i(t) = \text{sign}\Big(\sum_{k=1}^{N} U_{ik} x_k(t)\Big), \tag{7}$$

$$\nu_j(t) = H\Big(\kappa - x_j(t+1)\sum_{k=1}^{M} V_{jk} y_k(t)\Big), \tag{8}$$

$$V_{ji} \leftarrow V_{ji} + \eta\nu_j(t)x_j(t+1)y_i(t), \tag{9}$$

where $P_{ik}$ denotes the $(i, k)$ entry of the fixed random matrix $\mathbf{P}$, $H(\cdot)$ is the Heaviside function ($H(x) = 1$ if $x \geq 0$ and $H(x) = 0$ otherwise), $\kappa > 0$ is the robustness hyperparameter and $\eta > 0$ is the learning rate hyperparameter. $\mu_i(t)$ and $\nu_j(t)$ can be interpreted as the error terms for the hidden and the visible neurons, respectively. $z_i(t+1)$ can be interpreted as the synaptic input from an external neuron. The above procedure is then repeated for each $t$.

## 5.1 ANALYSIS

In this section, we provide theoretical analysis of the algorithm. The proofs are left to the Appendix. First, we provide convergence guarantee of the algorithm.

**Proposition 2** *Given the definitions in (4)(5)(7)(8), for all $i$, $j$ and $t$, if a solution exists such that $\mu_i(t) = 0$ and $\nu_j(t) = 0$ , then the algorithm (4)-(9) converges in finite steps given $U_{ij}$ and $V_{ji}$ are initialized to zero.*

Next, we show the algorithm can reduce error $\mu_i(t)$ for a single step of updating $\mathbf{U}$. The proposition can be trivially extended for $\nu_j(t)$ and $\mathbf{V}$ by a similar proof.

**Proposition 3** *Given the definitions in (4)(5), let*

$$U'_{ik} = U_{ik} + \eta\mu_i(t)z_i(t+1)x_k(t), \tag{10}$$

$$\mu'_i(t) = H\Big(\kappa - z_i(t+1)\sum_{k=1}^{N} U'_{ik} x_k(t)\Big). \tag{11}$$

*Then $\mu'_i(t) = 0$ for sufficiently large $\eta > 0$.*

To understand why reducing the errors $\mu_i(t)$ and $\nu_j(t)$ leads to sequence attractors, we present the following result.

**Proposition 4** *Given the definitions in (7)(8), let $\hat{\mathbf{y}}(t) = (\hat{y}_1(t), ..., \hat{y}_M(t)) \in \{-1, 1\}^M$ such that $\sum_k |\hat{y}_k(t) - y_k(t)| < \epsilon$. If $\nu_j(t) = 0$ and*

$$\epsilon \cdot \max_k |V_{jk}| < \kappa, \tag{12}$$

*then*

$$x_j(t+1) = \text{sign}\Big(\sum_{k=1}^{M} V_{jk}\hat{y}_k(t)\Big). \tag{13}$$

Proposition 4 shows that when the errors are zero, given perturbed hidden neuron states $\hat{\mathbf{y}}(t)$, we have $\mathbf{x}(t+1) = \text{sign}(\mathbf{V}\hat{\mathbf{y}}(t))$. The result can be trivially extended to show that given perturbed visible neuron states $\hat{\mathbf{x}}(t)$ we have $\mathbf{y}(t) = \text{sign}(\mathbf{U}\hat{\mathbf{x}}(t))$ by a similar proof. Therefore, the network can generate sequence $\mathbf{x}(1), ..., \mathbf{x}(T)$ as an attractor. From Proposition 4, we can also see that $\kappa$ acts as the robustness hyperparameter as it controls the level of perturbation $\epsilon$ for inequality (27) to hold.

To understand why the algorithm works despite that $\mathbf{P}$ is a random matrix and fixed during learning, consider the following. If the update of $\mathbf{U}$ converges, then $\mu_i(t) = 0$ for all $i$. Therefore,

$$y_i(t) = \text{sign}\Big( \sum_{k=1}^{N} U_{ik} x_k(t) \Big) = z_i(t+1) = \text{sign}\Big( \sum_{k=1}^{N} P_{ik} x_k(t+1) \Big). \tag{14}$$

The update of $\mathbf{V}$ aims at making the condition $\text{sign}\Big( \sum_{k=1}^{M} V_{jk} y_k(t) \Big) = x_j(t+1)$ hold, which is

$$\text{sign}(\mathbf{V}\text{sign}(\mathbf{P}\mathbf{x}(t+1))) = \mathbf{x}(t+1), \tag{15}$$

in matrix form when $y_k(t)$ is substituted by (14). For large $M$, a solution $\mathbf{V}$ exists, that is, the pseudo-inverse of $\mathbf{P}$ or the transpose of $\mathbf{P}$. The numerical result is shown in Figure 4. The phenomenon might be explained by the high-dimensional probability theory [Vershynin, 2018].

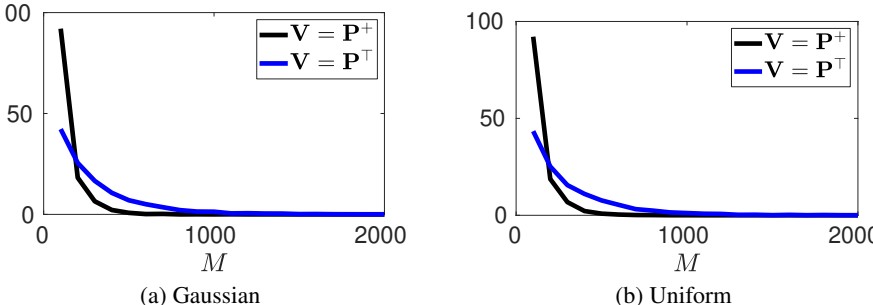

(a) Gaussian          (b) Uniform

Figure 4: Reconstruction error $\|\mathbf{x} - \text{sign}(\mathbf{V}\text{sign}(\mathbf{P}\mathbf{x}))\|_1$ where $\mathbf{P}$ is a $M \times N$ random matrix and $\mathbf{x}$ is a random vector uniformly sampled from $\{-1, 1\}^N$. $\mathbf{P}^+$ denotes the pseudo-inverse of $\mathbf{P}$. $\mathbf{P}^\top$ denotes the transpose of $\mathbf{P}$. (a) The entries of $\mathbf{P}$ are sampled i.i.d. from the standard Gaussian distribution. (b) The entries of $\mathbf{P}$ are sampled i.i.d. from the uniform distribution on $[-1, 1]$. In (a) and (b), $N = 100$ and the results are averaged over 100 trials. The results are similar in (a) and (b).

## 5.2 Robustness Hyperparameter

Having a hyperparameter $\kappa$ in the algorithm is not problematic in practice. One can simply set $\kappa = 1$ as we did for all the experiments in the next section and adjust the scale of initial weights and the learning rate. In margin perceptron, the margin learned is disproportional to the learning rate [Collobert & Bengio, 2004]. The margin is defined as the reciprocal of the weight magnitude, which is related to the robustness hyperparameter, as shown in Proposition 4. Therefore, it can be interpreted that the robustness hyperparameter is automatically adjusted during learning.

## 6 Experiments

We run experiments on synthetic and real-world sequence datasets for Hopfield networks with hidden neurons by the algorithm proposed in the previous section to learn sequence attractors. All the experiments are carried out in MATLAB and PyTorch. In all the experiments, each weight parameter of $\mathbf{U}$, $\mathbf{V}$ and $\mathbf{P}$ is sampled i.i.d. from Gaussian distribution with mean zero and variance $1 \times 10^{-6}$, learning rate $\eta = 1 \times 10^{-3}$ and robustness $\kappa = 1$. In each experiment, we run the algorithm for 500 epochs. In each epoch, the algorithm runs on $(\mathbf{x}(t), \mathbf{x}(t+1))$ from the start to the end of each sequence. No noise is added during learning. Noise is added only at retrieval. We also provide additional experiments in the Appendix for comparisons to networks of continuous neurons trained with backpropagation and to modern Hopfield networks for sequence learning [Chaudhry et al., 2023].

### 6.1 TOY EXAMPLES

To show the networks with hidden neurons can overcome the limitation of classical Hopfield networks, we conduct experiments on the examples in Figure 2. We construct a network of visible neurons $N = 10$ and hidden neurons $M = 50$ for each example. After learning, we test the robustness of the networks in retrieval by adding two salt-and-pepper noises (flipping the states of two out of ten neurons) to the first pattern of a sequence and set it to be the initial network state. The results are shown in Figure 5, from which we can see that the networks with hidden neurons can generate sequences which cannot be generated by classical Hopfield networks and retrieve them robustly under moderate level of noise.

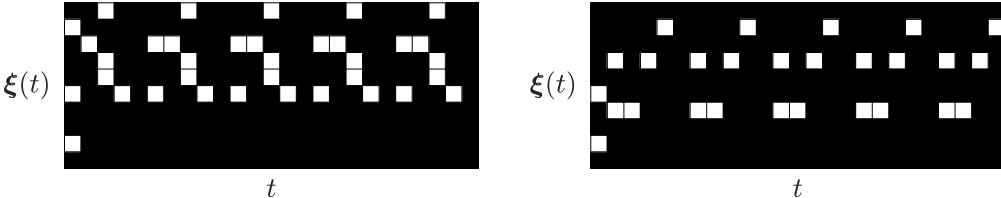

Figure 5: Networks with hidden neurons can generate the two sequences in Figure 2, which cannot be generated without hidden neurons, despite noisy initial states. Note that in the first column of each diagram two salt-and-pepper noises are added to test the robustness of the retrieval.

### 6.2 RANDOM SEQUENCES

We generate periodic sequences of random patterns $\mathbf{x}(1), ..., \mathbf{x}(T) \in \{-1, 1\}^N$. In each sequence, $\mathbf{x}(i) \neq \mathbf{x}(j)$ for $i \neq j$ except that $\mathbf{x}(1) = \mathbf{x}(T)$ for the periodicity. We set $N = 100$ and vary period length $T$. We sample each $\mathbf{x}(t)$ independently from the uniform distribution of $\{-1, 1\}^N$ for $t = 1, ..., T - 1$ and then resample it if it is identical to a previous pattern. Finally, we set $\mathbf{x}(T) = \mathbf{x}(1)$. For each random sequence, we construct a network with hidden neurons and apply the proposed learning algorithm. To evaluate the effectiveness of the learning algorithm, we compare learning only $\mathbf{V}$ (with $\mathbf{U}$ fixed during learning) and learning both $\mathbf{U}$ and $\mathbf{V}$. Once the learning is done, we test if the network can retrieve the sequence robustly given perturbed $\mathbf{x}(1)$ with 10 salt-and-pepper noises as the initial network state $\boldsymbol{\xi}(1)$. We define that the retrieval is successful if $\boldsymbol{\xi}(\tau + t) = \mathbf{x}(t)$ for some $\tau$ and all $t = 1, ..., T$. We run 100 trials for each $T$ or $M$ setting and count the successful retrievals.

In Figure 6, we show the visualization of a result. In Figure 7, we show the results with various period lengths $T$ for $M = 500$. In Figure 8, we show the results with various numbers of hidden neurons $M$ for $T = 70$. We can see learning both $\mathbf{U}$ and $\mathbf{V}$ is more effective than learning only $\mathbf{V}$. However, in both cases, the algorithm fails for large $T$, even if we increase the number of hidden neurons, which might be due to the suboptimality of the algorithm.

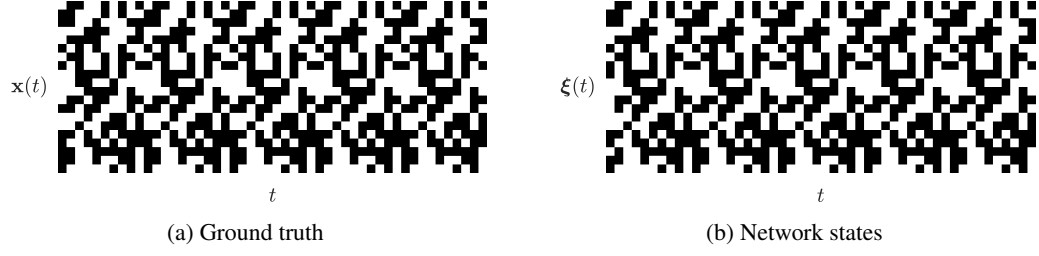

(a) Ground truth

(b) Network states

Figure 6: Learning random periodic sequences in networks with hidden neurons. Only the first 20 neurons of 100 visible neurons are selected for visualization due to space limitation. Note that in the first column of (b) salt-and-pepper noises are added to test the robustness of the retrieval.

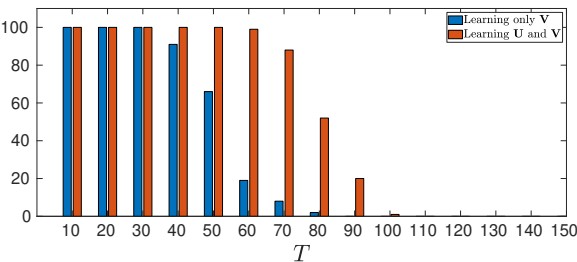

Figure 7: Successful retrievals out of 100 trials with different sequence period lengths.

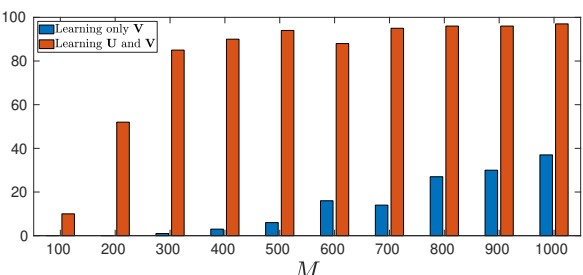

Figure 8: Successful retrievals out of 100 trials with different numbers of hidden neurons.

## 6.3 REAL-WORLD SEQUENCES

We test the networks with hidden neurons by the algorithm in learning real-world sequences on a silhouette sequence dataset (OU-ISIR gait database large population [Iwama et al., 2012]) and a handwriting sequence dataset (Moving MNIST [Srivastava et al., 2015]). The patterns in the sequences are rather correlated since adjacent image frames are similar. To adopt the datasets for the networks to learn, we convert the image intensity values to $\pm 1$. For the silhouette dataset, we use a network with hidden neuron number $M = 200$ to learn a single image sequence of length 103, in which each image has size $88 \times 128$. The images are flatten to vectors of size $88 \times 128 = 11264$. For the handwriting dataset, we use a network with hidden neuron number $M = 1000$ to learn 20 image sequences of length 20, in which each image has size $64 \times 64$. The images are flatten to vectors of size $64 \times 64 = 4096$. In Figure 9 and 10, we show the visualization results of the learned networks for robust retrieval, in which the first image of a sequence is corrupted and set to be the initial state of a network. In Figure 11, we show the average errors $\frac{1}{M} \sum_t \sum_i \mu_i(t)$ and $\frac{1}{N} \sum_t \sum_j \nu_j(t)$ during the learning process, from which we can see that both errors reduce to zero.

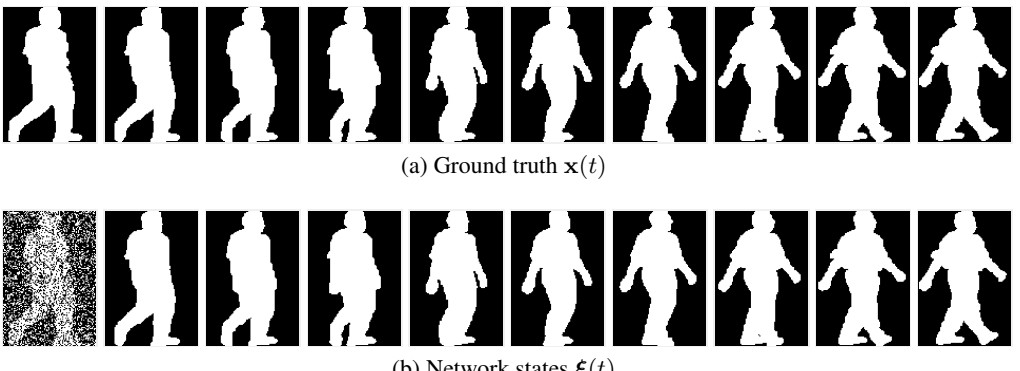

Figure 9: Retrieval of sequences under noise on the silhouette sequence dataset. An image sequence of length 103 is learned. Each image has size $88 \times 128$. In (a) and (b), $\mathbf{x}(t)$ and $\boldsymbol{\xi}(t)$ are shown respectively for $t = 1, ..., 10$. In (b), 2000 salt-and-pepper noises are added to the first image. The corrupted image is set to be the initial state of the network.

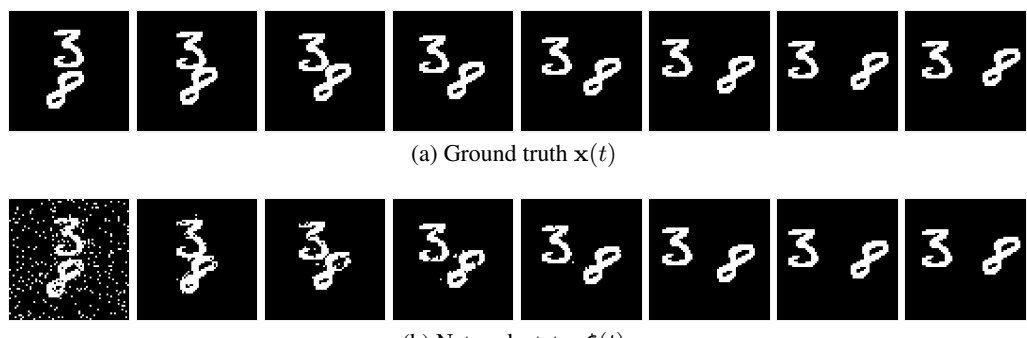

(a) Ground truth $\mathbf{x}(t)$

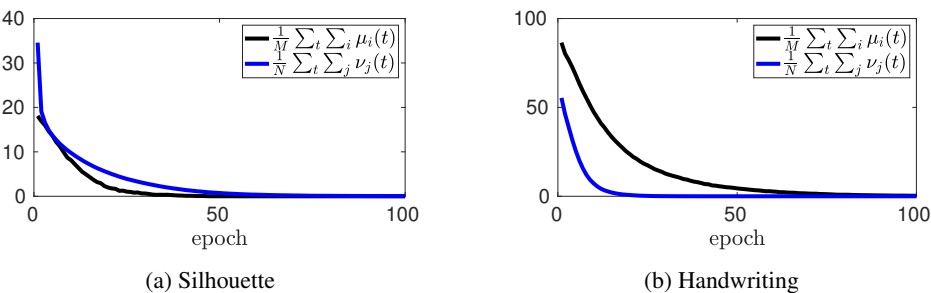

(b) Network states $\boldsymbol{\xi}(t)$

Figure 10: Retrieval of sequences under noise on the handwriting sequence dataset. 20 image sequences of length 20 are learned. Due to space limitation, only one image sequence is displayed in here. Each image has size $64 \times 64$. In (a) and (b), $\mathbf{x}(t)$ and $\boldsymbol{\xi}(t)$ are shown respectively for $t = 1, ..., 8$. In (b), 300 salt-and-pepper noises are added to the first image. The corrupted image is set to be the initial state of the network.

(a) Silhouette

(b) Handwriting

Figure 11: Errors during learning. $\frac{1}{M} \sum_t \sum_i \mu_i(t)$ is the average error for the hidden neurons. $\frac{1}{N} \sum_t \sum_j \nu_j(t)$ is the average error for the visible neurons.

## 7 CONCLUSION AND DISCUSSION

In this paper, we have investigated how networks of Hopfield type learn sequence attractors to represent temporal sequence information. We showed that to store arbitrary sequence patterns, it is necessary for the networks to include hidden neurons. We developed a local learning algorithm and demonstrated that our model works well on synthetic and real-world datasets. Thus, our work provides a possible biologically plausible mechanism in elucidating sequence memory in the brain.

In our model, hidden neurons are not directly involved in expressing pattern sequences. Instead, their contribution is on facilitating the storing and retrieving of pattern sequences. If the recurrent dynamics of our model is unfolded in time, the role of hidden neurons is analogous to that of a hidden layer in a two-layer feedforward network, where they serve as a relay stage for information processing. The indirect but indispensable role of hidden neurons may have a far-reaching implication to neural information processing.

In this paper, to pursue theoretical analysis, we have employed a very simple network model with binary neurons and threshold dynamics. From this simple model, we can get some insights into the neural mechanisms of sequence processing in the brain (as the classical Hopfield networks to static memories), but this simplification also incurs limitations, that is, to fully validate our results, further researches with biologically more plausible models are needed, which include, for instances, biologically more realistic neuron models, synapses models, connection structures, learning rules and form of pattern sequences. Additionally, external inputs, as an important part of temporal information processing in the brain, are absent in our network model and should be further investigated. These studies form our future work.

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

## A    COMPARISON TO NETWORKS OF CONTINUOUS NEURONS

We compare our networks of binary neurons (trained by the learning algorithm in Section 5) with networks of continuous neurons (trained with backpropagation) for learning the real-world sequences in Section 6.3. The states of the networks of continuous neurons are synchronously updated according to, for $i = 1, ..., M$ and $j = 1, ..., N$,

$$\zeta_i(t) = \tanh\Big( \sum_{k=1}^{N} U_{ik}\xi_k(t) \Big), \tag{16}$$

$$\xi_j(t+1) = \tanh\Big( \sum_{k=1}^{M} V_{jk}\zeta_k(t) \Big), \tag{17}$$

which can be interpreted as the continuous approximation of the networks in (2)(3).

We follow the experimental settings of Section 6.3. In training the networks of continuous neurons, we compared two algorithms, stochastic gradient descent (SGD) and Adam. We used the default weight initialization in PyTorch. We tested learning rates $\{1 \times 10^{-1}, 1 \times 10^{-2}, 1 \times 10^{-3}, 1 \times 10^{-4}, 1 \times 10^{-5}, 1 \times 10^{-6}\}$ in each set of experiments and show the best performance in terms of training speed in Figure 12. The training error is defined as $\sum_t \|\mathbf{x}(t+1) - \phi(\mathbf{V}\phi(\mathbf{U}\mathbf{x}(t)))\|_1$ where $\phi$ is the sign or $\tanh$ function. We found that networks of continuous neurons can reach zero training errors and are robust to noise in sequence retrieval as ours. However, our network model (2)(3), together with our algorithm, learns much faster compared to networks of continuous neurons trained with backpropagation, as shown in Figure 12.

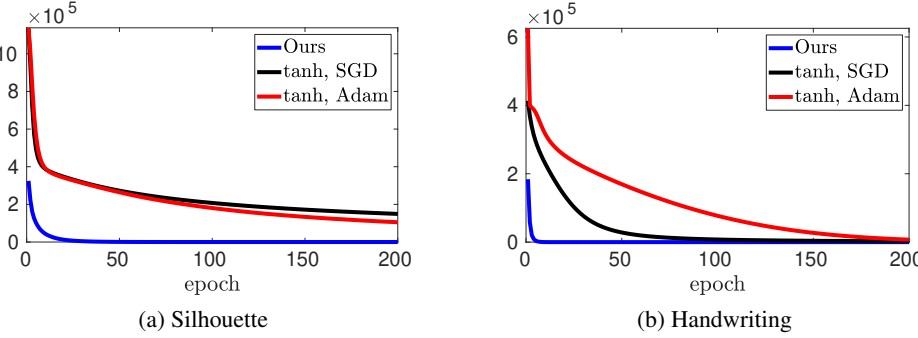

(a) Silhouette                    (b) Handwriting

Figure 12: Training errors

We summarize some advantages of our approach compared to networks of continuous neurons trained with backpropagation.

- Our algorithm is more biologically plausible, as it requires only local information between neurons, while networks of continuous neurons requires backpropagation, whose biological plausibility is in question.
- Our algorithm is proven to converge and lead to sequence attractors. Such theoretical results are missing for networks of continuous neurons trained with backpropagation.
- Our algorithm converges to zero training error much faster empirically.

## B    COMPARISON TO MODERN HOPFIELD NETWORKS

Modern Hopfield networks can also be interpreted as networks with hidden neurons [Krotov & Hopfield, 2016; Chaudhry et al., 2023]. Therefore, we conducted experiments with modern Hopfield networks for learning the real-world sequences in Section 6.3. We used the model and the learning procedure in [Chaudhry et al., 2023]. We found modern Hopfield networks can indeed learn to store the sequences in Section 6.3 and retrieve them robustly under noise if we used the nonlinear activation function $f(x) = x^d$ in the model with $d \geq 30$.

We summarize some advantages of our approach compared to modern Hopfield networks.

- Our network model is more biologically plausible as it requires only the threshold activation function, while modern Hopfield networks require a high-order polynomial or exponential activation function, whose biological plausibility is less clear. Despite that some efforts have been made in [Chaudhry et al., 2023] for a biologically justification, our network model is more natural to interpret.
- Our algorithm is more biologically plausible as it requires only local information between neurons, while modern Hopfield networks require the pseudo-inverse learning algorithm, whose biological plausibility is in question. Besides, our algorithm learns the sequences in an online fashion while the learning procedure in [Chaudhry et al., 2023] is a batch-based method, which is again biologically less plausible.

In [Chaudhry et al., 2023], it has been shown that modern Hopfield networks can store long sequences, for a sufficient number of hidden neurons, This is also possible with our network model based on an explicit construction in the proof of Proposition 1.

## C  PROOF OF PROPOSITION 1

We construct a network such that, given $\boldsymbol{\xi}(t) = \mathbf{x}(i)$ for $i = 1, ..., T-1$, the hidden neurons provide an one-hot encoding of the successive pattern $\mathbf{x}(i+1)$, which is then decoded to be $\boldsymbol{\xi}(t+1)$.

To store $\mathbf{x}(1), ..., \mathbf{x}(T) \in \{-1, 1\}^N$ in (2)(3), assuming $\mathbf{x}(i) \neq \mathbf{x}(j)$ for $i \neq j$ except that $\mathbf{x}(1) = \mathbf{x}(T)$, let $M = T - 1$ and construct weight matrix $\mathbf{U}$ as

$$\mathbf{U} = (\mathbf{x}(1), \mathbf{x}(2), ..., \mathbf{x}(T-1))^\top \tag{18}$$

and hidden neurons $\boldsymbol{\zeta}(t) = (\zeta_1(t), ..., \zeta_M(t))$ as

$$\zeta_i(t) = \text{sign}\Big( \sum_{k=1}^N U_{ik}\xi_k(t) - N \Big) \tag{19}$$

$$= \text{sign}\big( \mathbf{x}(i)^\top \boldsymbol{\xi}(t) - N \big) \tag{20}$$

such that given $\boldsymbol{\xi}(t) = \mathbf{x}(i)$ for $i = 1, ..., T-1$, we have

$$\zeta_j(t) = \begin{cases} +1, & \text{if } j = i, \\ -1, & \text{otherwise.} \end{cases} \tag{21}$$

Next, we construct the weight matrix $\mathbf{V}$ as

$$\mathbf{V} = (\mathbf{x}(2), \mathbf{x}(3), ..., \mathbf{x}(T)) \tag{22}$$

and visible neurons $\boldsymbol{\xi}(t+1) = (\xi_1(t+1), ..., \xi_N(t+1))$ as

$$\boldsymbol{\xi}(t+1) = \text{sign}(\mathbf{V}\boldsymbol{\zeta}(t) + \boldsymbol{\theta}) \tag{23}$$

where $\boldsymbol{\theta} = \sum_{j=2}^T \mathbf{x}(j)$ such that given the one-hot vector $\boldsymbol{\zeta}(t)$ we have

$$\boldsymbol{\xi}(t+1) = \text{sign}\Big( \mathbf{x}(i+1) - \sum_{j \neq i+1} \mathbf{x}(j) + \sum_{j=2}^T \mathbf{x}(j) \Big) \tag{24}$$

$$= \text{sign}(2 \cdot \mathbf{x}(i+1)) \tag{25}$$

$$= \mathbf{x}(i+1) \tag{26}$$

## D  PROOF OF PROPOSITION 2

Note that the update of $\mathbf{U}$ (4)(5)(6) in Section 5 does not depend on $\mathbf{V}$. Therefore, we first prove the convergence of updating $\mathbf{U}$ for $\eta > 0$ and $\kappa > 0$. The proof follows from [Gardner, 1988]. Assume $\mathbf{U}^*$ exists such that, for all $t$ and $i$,

$$z_i(t+1) \sum_k U_{ik}^* x_k(t) \geq \kappa. \tag{27}$$

Define the $p$-th update of $\mathbf{U}$ with $\mu_i(t_p) = 1$ by

$$U_{ij}^{(p+1)} = U_{ij}^{(p)} + \eta z_i(t_p + 1) x_j(t_p) \tag{28}$$

for some $t_p \in \{1, ..., T-1\}$ and all $j$ in parallel. We assume zero-initialization, that is, $U_{ij}^{(1)} = 0$ for simplicity but the result holds if $|U_{ij}^{(1)}|$ is sufficiently small. Let

$$X_i^{(p+1)} = \frac{\sum_j U_{ij}^{(p+1)} U_{ij}^*}{\sqrt{\sum_j \left(U_{ij}^{(p+1)}\right)^2} \sqrt{\sum_j \left(U_{ij}^*\right)^2}}. \tag{29}$$

The Cauchy-Schwarz inequality, we have $X_i^{(p+1)} \le 1$. Now we prove the convergence of updating $\mathbf{U}$ by contradiction. Assuming the update of $\mathbf{U}$ does not converge, we will show that $X_i^{(p+1)} > 1$ as $p \to \infty$. First, we have

$$\sum_j U_{ij}^{(p+1)} U_{ij}^* - \sum_j U_{ij}^{(p)} U_{ij}^* = \eta \sum_j z_i(t_p + 1) U_{ij}^* x_j(t_p) \ge \eta \kappa \tag{30}$$

due to (27) and therefore

$$\sum_j U_{ij}^{(p+1)} U_{ij}^* = \sum_j U_{ij}^{(p+1)} U_{ij}^* - \sum_j U_{ij}^{(p)} U_{ij}^* + ... + \sum_j U_{ij}^{(2)} U_{ij}^* - \sum_j U_{ij}^{(1)} U_{ij}^* + \sum_j U_{ij}^{(1)} U_{ij}^* \tag{31}$$

$$\ge \eta \kappa p \tag{32}$$

since we assumed $U_{ij}^{(1)} = 0$. Next, we have

$$\sum_j \left(U_{ij}^{(p+1)}\right)^2 - \sum_j \left(U_{ij}^{(p)}\right)^2 = \sum_j \left(U_{ij}^{(p)} + \eta z_i(t_p+1) x_j(t_p)\right)^2 - \sum_j \left(U_{ij}^{(p)}\right)^2 \tag{33}$$

$$= 2\eta \sum_j U_{ij}^{(p)} z_i(t_p+1) x_j(t_p) + N\eta^2 \tag{34}$$

$$= 2\eta z_i(t_p+1) \sum_j U_{ij}^{(p)} x_j(t_p) + N\eta^2 \tag{35}$$

$$< 2\eta \kappa + N\eta^2 \tag{36}$$

since we assumed $\mu_i(t_p) = 1$ and therefore $z_i(t_p+1) \sum_j U_{ij}^{(p)} x_j(t_p) < \kappa$. Then, we have

$$\sqrt{\sum_j (U_{ij}^{(p+1)})^2} - \sqrt{\sum_j (U_{ij}^{(p)})^2} \tag{37}$$

$$= \left(\sum_j \left(U_{ij}^{(p+1)}\right)^2 - \sum_j \left(U_{ij}^{(p)}\right)^2\right) \Big/ \left(\sqrt{\sum_j \left(U_{ij}^{(p+1)}\right)^2} + \sqrt{\sum_j \left(U_{ij}^{(p)}\right)^2}\right) \tag{38}$$

$$< (2\eta\kappa + N\eta^2) \Big/ \left(\sqrt{\sum_j \left(U_{ij}^{(p+1)}\right)^2} + \sqrt{\sum_j \left(U_{ij}^{(p)}\right)^2}\right). \tag{39}$$

By Cauchy-Schwarz inequality, we have

$$\sqrt{\sum_j (U_{ij}^{(p+1)})^2} \sqrt{\sum_j (U_{ij}^*)^2} \ge \sum_j U_{ij}^{(p+1)} U_{ij}^* \ge \eta \kappa p \tag{40}$$

and therefore

$$\sqrt{\sum_j (U_{ij}^{(p+1)})^2} \ge \frac{\eta \kappa p}{\sqrt{\sum_j (U_{ij}^*)^2}}. \tag{41}$$

Also,

$$\sqrt{\sum_j (U_{ij}^{(p+1)})^2} = \sqrt{\sum_j (U_{ij}^{(p+1)})^2} - \sqrt{\sum_j (U_{ij}^{(p)})^2} + ... + \sqrt{\sum_j (U_{ij}^{(2)})^2} - \sqrt{\sum_j (U_{ij}^{(1)})^2} \quad (42)$$

$$+ \sqrt{\sum_j (U_{ij}^{(1)})^2} \quad (43)$$

$$< \sum_{q=1}^p (2\eta\kappa + N\eta^2) \Big/ \left( \sqrt{\sum_j (U_{ij}^{(q+1)})^2} + \sqrt{\sum_j (U_{ij}^{(q)})^2} \right) \quad (44)$$

$$< \sum_{q=1}^p (2\eta\kappa + N\eta^2) \sqrt{\sum_j (U_{ij}^*)^2} \frac{1}{\eta\kappa(2q-1)} \quad (45)$$

$$= \frac{\eta\kappa + N\eta^2/2}{\eta\kappa} \sqrt{\sum_j (U_{ij}^*)^2} \sum_{q=1}^p \frac{1}{q-1/2} \quad (46)$$

due to (39) and $U_{ij}^{(1)} = 0$. Note that for $q > 1$

$$\frac{1}{q-1/2} \le \int_{q-3/2}^{q-1/2} \frac{1}{x} dx = \log(q-1/2) - \log(q-3/2) \quad (47)$$

and

$$\sum_{q=1}^p \frac{1}{q-1/2} = \frac{1}{2} + \sum_{q=2}^p \frac{1}{q-1/2} \le \frac{1}{2} + \int_{1/2}^{p-1/2} \frac{1}{x} dx = 2 + \log(p-1/2) - \log(1/2). \quad (48)$$

Therefore,

$$\sqrt{\sum_j (U_{ij}^{(p+1)})^2} = O(\log(p)) \quad (49)$$

and

$$\sum_j U_{ij}^{(p+1)} U_{ij}^* = \Omega(p) \quad (50)$$

as $p \to \infty$. We have,

$$X_i^{(p+1)} = \frac{\sum_j U_{ij}^{(p+1)} U_{ij}^*}{\sqrt{\sum_j (U_{ij}^{(p+1)})^2} \sqrt{\sum_j (U_{ij}^*)^2}} > 1 \quad (51)$$

for some $p$. This contradicts that $X_i^{(p+1)} \le 1$. Thus, the updating $\mathbf{U}$ converges.

Upon the convergence of updating $\mathbf{U}$, we can prove the convergence of $\mathbf{V}$ if there exists $\mathbf{V}^*$ such that for all $t$ and $i$,

$$x_i(t+1) \sum_k V_{ik}^* y_k(t) \ge \kappa \quad (52)$$

by a similar proof.

# E   PROOF OF PROPOSITION 3

If $\mu_i(t) = 0$, then $U'_{ik} = U_{ik}$ and $\mu'_i(t) = \mu_i(t) = 0$. If $\mu_i(t) = 1$, then

$$\mu'_i(t) = H\left(\kappa - z_i(t+1)\sum_{k=1}^{N}\left(U_{ik} + \eta z_i(t+1)x_k(t)\right)x_k(t)\right) \tag{53}$$

$$= H\left(\kappa - z_i(t+1)\sum_{k=1}^{N}U_{ik}x_k(t) - \eta\left(z_i(t+1)\right)^2\sum_{k=1}^{N}\left(x_k(t)\right)^2\right) \tag{54}$$

$$= H\left(\kappa - z_i(t+1)\sum_{k=1}^{N}U_{ik}x_k(t) - \eta N\right) = 0 \tag{55}$$

for sufficiently large $\eta > 0$ given $x_k(t) = \pm 1$, $z_i(t+1) = \pm 1$ and the property of Heaviside function.

# F   PROOF OF PROPOSITION 4

If $\nu_j(t) = 0$, then we have

$$x_j(t+1)\sum_{k=1}^{M}V_{jk}y_k(t) \geq \kappa. \tag{56}$$

Next,

$$x_j(t+1)\sum_{k=1}^{M}V_{jk}\hat{y}_k(t) = x_j(t+1)\sum_{k=1}^{M}V_{jk}\left(y_k(t) + \hat{y}_k(t) - y_k(t)\right) \tag{57}$$

$$= x_j(t+1)\sum_{k=1}^{M}V_{jk}y_k(t) + x_j(t+1)\sum_{k=1}^{M}V_{jk}\left(\hat{y}_k(t) - y_k(t)\right) \tag{58}$$

$$\geq \kappa + x_j(t+1)\sum_{k=1}^{M}V_{jk}\left(\hat{y}_k(t) - y_k(t)\right) \tag{59}$$

$$\geq \kappa - \left|\sum_{k=1}^{M}V_{jk}\left(\hat{y}_k(t) - y_k(t)\right)\right| \tag{60}$$

$$\geq \kappa - \max_k|V_{jk}|\sum_{k=1}^{M}|\hat{y}_k(t) - y_k(t)| \tag{61}$$

$$> \kappa - \max_k|V_{jk}| \cdot \epsilon > 0 \tag{62}$$

since $x_j(t+1) = \pm 1$, which implies

$$x_j(t+1) = \text{sign}\left(\sum_{k=1}^{M}V_{jk}\hat{y}_k(t)\right). \tag{63}$$

