# OpenReview forum: "Learning Sequence Attractors in Recurrent Networks with Hidden Neurons"
_ICLR.cc/2024/Conference — Submitted to ICLR 2024_

### Official Review · Reviewer_qcYh · 2023-10-29

**Soundness:** 3 good
**Presentation:** 2 fair
**Contribution:** 2 fair
**Rating:** 5
**Confidence:** 5

**Summary:**

This paper introduces an algorithm to let a recurrent neural network learn sequences of patterns with convergence guarantees for the algorithm.
Furthermore, the importance of hidden neurons in the proposed architecture and activation function is shown for producing some of the sequences.

**Strengths:**

The effectiveness of the approach is demonstrated against two datasets, highlighting its applicability. The theorems and claims are sound.
Furthermore, the paper is well written and the contributions are clearly explained with possible implications of the work for neuroscience. These implications for neuroscience might provide a new perspective on the contribution of some neurons to neural computation.

**Weaknesses:**

\paragraph{Comparison of encoding efficiency to other methods/experiments}
The authors don't make considerable efforts to compare their work to previous ones relating to sequence encoding.
There are many works on sequence learning, see for example (Lipton, 2015). Your work is a particular subclass of networks that can perform sequence-to-sequence processing (namely where you have a sequence with an single input and the start and zero inputs afterwards). Make comparison with other methods to measure the performance of your framework in terms of robustness to noise, memory capacity (number of recallable pattern sequences), etc.

\paragraph{Other experiments to demonstrate the effectiveness of the method}
The the demonstration contribution of this framework could benefit from some additional experiments.
Experiments for effect of noise level of the first patter on the performance.
Further, the paper doesn't address how inputs to the network influence the output or performance. Experiments that track performance of the networks as a function of injected noise during sequence generation would better demonstrate the usefulness of this framework for neuroscience.


\paragraph{Figures}
Overall, the information in the figures is very low.
For Figure 4 and 11 for example it is unclear what is to be gained from seeing the convergence of the algorithm, if it is not compared to other algorithms to see which converges in less epochs for example.

For Figure 7 and 8 it would be beneficial to see how increasing $T$ influences retrieval success as $M$ and $N$ are changed.

\paragraph{Some remarks on concepts and notation}
The section on the robustness hyperparameter is unclear. Better describe how the margin of the margin perceptron is related to the robustness here.

Relating to the derivations:Why does such a $U^\star$ exist?
After Equation (19): "due to (1)". Should this be "due to (17)"?

There are a couple of unintroduced variables/notation.
In Eq. (33) $q$ is appearing without introduction. Should this be $p$? Clarify step (32) to (33) in the derivation.
In Eq. (39) $\Omega$ has not been introduced. Did you mean $O$?


\paragraph{Neuroscience implications}
The justification of the relevance of this work to V1 neurons is insufficient.
The authors should justify why this particular activation function is a good model for V1 neural activity. Furthermore, the claim about unexplained V1 neural activity relies on the limit of the Heaviside activation function, for other activation functions hidden neurons would not be necessary.
But more fundamentally, the authors seem to understand V1 neural activity dynamics in a very different way than the general con census. Although there is sufficient recurrence in V1, the main function of these neurons is not sequence generation. Hippocampal circuits (that are mentioned in the beginning of the paper) would be a better justification.
Finally, for V1 neurons in particular, but for all subnetworks in the brain really, inputs are a very important contribution to the dynamics.


Lipton, Z.C., Berkowitz, J. and Elkan, C., 2015. A critical review of recurrent neural networks for sequence learning. arXiv preprint arXiv:1506.00019.

**Questions:**

What is the expressivity of the Heaviside function for a fixed $M$ and $N$ in terms of $T$?

On page 8, just above Figure 4, when is $M$ large enough? When is the solution the pseudo-inverse and when the transpose of $\mathbf{P}$? How can high-dimensional probability theory explain this?

---

> ### Author Response · Authors · 2023-11-18
>
> We thank the reviewer for the dedicated proofreading and helpful suggestions on improving our paper. Below, we address each point of yours.
>
> * **Comparison of encoding efficiency to other methods/experiments**
>     In the revised paper, we have compared our approach with networks of continuous neurons trained with backpropagation and modern Hopfield networks. Please see Appendix A and B in the revised paper.
>
> * **Experiments for effect of noise level of the first patter on the performance.**
>     We observed a trade-off between the noise level that can be tolerated and the length of the stored sequence in our experiments. Additionally, we provide the code of our experiments in the supplementary material for further exploration.
>
> * **Further, the paper doesn't address how inputs to the network influence the output or performance.**
>     We agree that the inputs are important. However, in this paper, we focus on spontaneous neural activities and sequence memory for which the inputs are absent. The exploration of inputs to the network is beyond the scope of this paper.
>
> * **For Figure 4 and 11 for example it is unclear what is to be gained from seeing the convergence of the algorithm, if it is not compared to other algorithms to see which converges in less epochs for example.**
>     Figure 4 is to demonstrate that why the algorithm works despite that $\mathbf{P}$ is a random matrix and fixed during learning. Figure 11 is to demonstrate that both errors $\mu_i(t)$ and $\nu_j(t)$ would jointly reduce to zero, rather than conflicting with each other. Additionally, we provide comparison experiments in Figure 12 in Appendix A.
>
> * **For Figure 7 and 8 it would be beneficial to see how increasing  influences retrieval success as and are changed.**
>     Due to space limitation, we cannot provide all possible experiments suggested by the reviewer in the paper, as the number of combinations of the hyperparameters is enormous. Nevertheless, we provide the code of our experiments in the supplementary material for further exploration.
>
> * **Better describe how the margin of the margin perceptron is related to the robustness here.**
>     In margin perceptron, the margin is disproportional to the magntitude of the weights, which is related to robustness, as can be been in eq (12) in Proposition 4. We improved the presentation in the revised paper.
>
> * **Why does such a** $\mathbf{U}^*$ **exist?**
>     The assumption that such a $\mathbf{U}^*$ exists is a condition that the algorithm converges in finite steps. This condition is also in convergence proof of the perceptron learning algorithm.
>
> * **After Equation (19): "due to (1)". Should this be "due to (17)"?**
>     We corrected this typo in the revised paper.
>
> * **In Eq. (33) $q$ is appearing without introduction**
>     $q$ is an index which ranges from $1$ to $p$. It should be clear from the context.
>
> * **Clarify step (32) to (33) in the derivation.**
>     We improved our writing of the derivation in the revised paper.
>
> * **In Eq. (39) $\Omega$ has not been introduced.**
>     $\Omega$ (Big-Omega) is a standard notation in for analyzing complexity of algorithms. It is defined as the asymptotic lower bounds.
>
> * **The justification of the relevance of this work to V1 neurons is insufficient.**
>     The reviewer made good points on our speculation on V1. After reconsideration, we found our speculation on V1 is too premature and therefore removed it in the revised paper.
>
> * **What is the expressivity of the Heaviside function for a fixed $M$ and $N$ in terms of $T$?**
>     In the revised paper, we derived the lower bound of the capacity of our network model. The maximal length of sequences that can be stored is at least $M$ (number of hidden neurons).  Please see the updated Proposition 1 in the revised paper.
>
> * **On page 8, just above Figure 4, when is $M$
>  large enough?**
>  Empirically, we found when $M$ is slightly larger than the sequence length $T$, the network can learn to store the sequence. Interestingly, we also found that larger $M$ leads to faster convergence in learning.
>
> * **When is the solution the pseudo-inverse and when the transpose of $P$?**
>  Both works fine when $M$ is large enough as shown in Figure 4.
>
> * **How can high-dimensional probability theory explain this?**
> The phenomenon that $\mathbf{x} = \text{sign}(\mathbf{V} \text{sign}(\mathbf{Px}))$ for $\mathbf{V}=\mathbf{P}^\top$ or $\mathbf{V}=\mathbf{P}^+$ only happens when $M$ is sufficiently large. We cannot prove it at the moment but offer our intuition and conjecture for the interested reader to explore further.

---

> > ### Comment · Reviewer_qcYh · 2023-11-22
> >
> > Thank you for the response. I appreciate the additional details and the new version is taking steps in the right direction.
> >
> > > Comparison of encoding efficiency to other methods/experiments
> >
> > It seems like the trained neural networks for the comparison are feedforward ones. Why did the authors not choose RNNs, which would also make the comparison more biologically relevant.
> >
> > >> Our algorithm is proven to converge and lead to sequence attractors. Such theoretical results are missing for networks of continuous neurons trained with backpropagation
> >
> > As RNNs have universal approximation properties (at least for such tasks as considered in the paper), the two frameworks seem to have the same expressivity for the limit where the number of neurons go to infinity.
> >
> > Finally, being able to respond to incoming stimuli is an important aspect of biological networks, and it is possible to account for with RNNs, which is missing for the proposed framework.
> >
> > > Further, the paper doesn't address how inputs to the network influence the output or performance.
> >
> > For the neuroscientific relevance having experiments that measure sequence memory ability in the presence of continuously incoming stimuli or noise would have been essential.
> >
> > > Proposition 1
> >
> > This seems like an inefficient way to encode sequences by using almost the number of neurons for the sequence length. Also, this is considering a single sequence, which limits the applicability of this result.
> >
> >
> > > The justification of the relevance of this work to V1 neurons is insufficient.
> >
> > Without this neuroscientific justification, the proposed framework is missing a clear contribution for neuroscience.

---

> ### Author Response · Authors · 2023-11-23
>
> > It seems like the trained neural networks for the comparison are feedforward ones. Why did the authors not choose RNNs, which would also make the comparison more biologically relevant.
>
> Without inputs, RNN is equivalent to a feedforward network when it is unfolded in time. Please see Figure 1 and 3.
>
> > Finally, being able to respond to incoming stimuli is an important aspect of biological networks, and it is possible to account for with RNNs, which is missing for the proposed framework.
>
> We focus on spontaneous neural activities and sequence memory for which the inputs are absent. Spontaneous or internally generated activities have been observed in hippocampus [Pastalkova, et al. 2008]. Our comparison work does not consider inputs either [Amari 1972, Bressloff and Taylor 1992, Chaudhry et al., 2023].
>
> > This seems like an inefficient way to encode sequences by using almost the number of neurons for the sequence length.
>
> Encoding sequences by using almost the number of neurons for the sequence length also appears in our comparison work [Chaudhry et al., 2023]. In the future, when considering multi-layer hidden neurons, we might be able to derive more efficient capacity bound.
>
> > Also, this is considering a single sequence, which limits the applicability of this result.
>
> One can concatenate multiple sequences into a single sequence as in our comparison work [Chaudhry et al., 2023].
>
> > Without this neuroscientific justification, the proposed framework is missing a clear contribution for neuroscience.
>
> As discussed in the Abstract and Introduction, it remains largely unclear how the brain learns to store and retrieve sequence memories. Our work provides a possible biologically plausible mechanism in elucidating sequence memory in the brain. We have added this in the revised paper. Thanks for the suggestion.
>
>
> References
>
> Internally generated cell assembly sequences in the rat hippocampus. Pastalkova, et al. Science, 2008.
>
> Learning patterns and pattern sequences by self-organizing nets of threshold elements. Amari. IEEE Transactions on Computers, 1972.
>
> Perceptron-like learning in time-summating neural networks. Bressloff and Taylor. Journal of Physics A: Mathematical and General, 1992.
>
> Long sequence hopfield memory. Chaudhry et al, NeurIPS, 2023.

---

> ### Author Response · Authors · 2023-11-23
> **External inputs**
>
> We agree with the reviewer that external inputs are crucial in neural information processing. However, it is beyond the scope of this paper. We have acknowledged this limitation in the Conclusion and Discussion section of the revised paper and plan to explore this important topic in the future work.

---

### Official Review · Reviewer_TBVF · 2023-10-31

**Soundness:** 2 fair
**Presentation:** 2 fair
**Contribution:** 2 fair
**Rating:** 5
**Confidence:** 4

**Summary:**

Attractor networks can be considered all-to-all connected feedforward networks without any hidden layer. These ‘visible neurons’ must do both the computational and representational work simultaneously. This limits the expressibility of such networks, particularly for sequence attractors. This submission shows one way around this limitation is to add hidden neurons with a dedicated computational role and no (direct) representational role. A learning rule is introduced to learn the necessary parameters for these hidden neurons in the case of artificial and naturalistic data, with good recall shown.

**Strengths:**

Integrates some nice ideas to solve an identified problem.

**Weaknesses:**

*Novelty*

I question how or in what way this is new or different to past work (see question 2 below), and for that reason am concerned about novelty. However, my concern might be misplaced and I would appreciate the authors clarifying this point.

*Imprecise or incorrect statement*

From page 1: “the Hopfield model and related works typically only consider static attractors”

No, there exist many works looking at non-static attractors. Some of these are cited at the end of the first sentence in section 2 on page 2. A few additional examples include:

H. Gutfreund and M. Mezard. Processing of temporal sequences in neural networks. Phys. Rev. Lett., 61:235–238 1998.

Arjun Karuvally, Terrence Sejnowski, and Hava T Siegelmann. General sequential episodic memory model, ICML 2023.

Hamza Chaudhry, Jacob Zavatone-Veth, Dmitry Krotov, Cengiz Pehlevan, Long Sequence Hopfield Memory, NeurIPS 2023.

*Lack of comparisons and practical applications*

Previous work on encoding sequences in attractor networks have taken many different approaches. Additionally, there exist many methods for learning sequences in the machine learning literature. There is a lack of comparison with these prior methods.

**Questions:**

1. Where is the evidence that the examples shown in Figure 2 cannot be generated by a network without hidden neurons?

2. How is your approach conceptually different to hierarchical attractor network architectures? E.g.,

Dmitry Krotov, Hierarchical Associative Memory, arXiv:2107.06446

Kunihiko Fukushima, A hierarchical neural network model for associative memory, Biological Cybernetics volume 50, pages105–113 (1984)

3. What is the memory capacity of this network? How does this depend on memory load?

4. In the second paragraph of the conclusion, there is some speculation that neurons in V1 with unknown function may be akin to the hidden neurons of this model. How would a neuroscientist test for this? How should the tuning properties be studied, i.e., what should be measured? Does there exist some structure(s) in the hidden neuron activity data from your own model which you would expect in the aforementioned V1 neurons?

---

> ### Author Response · Authors · 2023-11-18
>
> We thank the reviewer for the feedback and helpful suggestions on improving our paper. Below, we address each point of yours.
>
> * **No, there exist many works looking at non-static attractors.**
>     What we meant is most previous works on attractor networks of Hopfield type considered only static attractors. Although there are also some other works on sequence attractors as the reviewer pointed out and we cited, the amount of them is smaller compared to those on static attractors. We have clarified this in the revised paper.
>
> * **There is a lack of comparison with these prior methods.**
> In the revised paper, we have compared our approach with networks of continuous neurons trained with backpropagation and modern Hopfield networks. Please see Appendix A and B in the revised paper.
>
> * **Where is the evidence that the examples shown in Figure 2 cannot be generated by a network without hidden neurons?**
> "In Figure 2, we show additional examples of
> sequences that cannot be generated by the network. The sequences are synthetically constructed. We
> then test if the perceptron learning algorithm can learn the sequences. Since the algorithm converges
> if the linear separability condition is met [Minsky \& Papert, 1969], the divergence of the algorithm
> implies that the sequences cannot be generated by the networks." This is stated in the text above Figure 2.
>
> * **How is your approach conceptually different to hierarchical attractor network architectures?**
> We do not claim our approach is conceptually different to hierarchical attractor network architectures such as [Chaudhry et al, NeurIPS 2023] for sequence learning. However, our model and algorithm are more biologically plausible, as stated in Appendix B in the revised paper.
>
> * **What is the memory capacity of this network? How does this depend on memory load?**
> In the revised paper, we derived the lower bound of the capacity of our network model. The maximal length of sequences that can be stored is at least $M$ (number of hidden neurons).  Please see the updated Proposition 1 in the revised paper.
>
> * **In the second paragraph of the conclusion, there is some speculation that neurons in V1 with unknown function may be akin to the hidden neurons of this model.**
> The reviewer raised good questions for us to reflect on our speculation. After reconsideration, we found our speculation on V1 is too premature and therefore removed it in the revised paper.

---

### Official Review · Reviewer_9sz4 · 2023-11-01

**Soundness:** 3 good
**Presentation:** 2 fair
**Contribution:** 4 excellent
**Rating:** 8
**Confidence:** 4

**Summary:**

In this paper, the authors extend Hopfield networks by adding one hidden layer and together with their proposed learning rules, show that this architecture can store and retrieve binary sequences. They provide convergence guarantees and empirically demonstrate, on various toy and real world examples, that their network learns to store sequences and retrieve them even in the presence of (a particular type of) noise.

**Strengths:**

The paper provides a relatively simple extension to Hopfield networks and corresponding learning rules to store and retrieve sequences that works well, which is novel to my knowledge. The problem of being able to store and retrieve sequences is an important one, both for understanding biological brains and for machine learning. Therefore having such a method is very useful and significant for the community.

The paper provides convergence proofs and empirical evaluation on a variety of targeted and relevant tasks, which also makes the paper a high quality contribution to the community. I particularly found the specific examples provided to demonstrate the problems with storing sequences in a fully visible recurrent network very useful to understand the motivation behind their approach.

**Weaknesses:**

A discussion of the capacity of the proposed architecture is missing, and is pretty important to be able to meaningfully connect this approach with biology and apply it in machine learning. This, in my opinion, is the biggest weakness of the paper.

The experiments section also does not have sufficient details about hyper-parameters ($
\eta, \tau$). The language and clarity of the exposition in the paper could be improved significantly.  See examples in "Questions".

**Questions:**

## Questions related to points mentioned above:
- How many times is each sequence presented to the model?
- What's the capacity of the model? How many sequences can it store?

## Clarity issues in the paper:
- The sign function and the Heaviside function seem to be identical the way it's been defined in the paper.
- The first sentence of Related work just lists a bunch of papers, which seems redundant, since these papers are explained later anyway.

### Minor:
Would have been useful to have Fig. 2 and Fig. 5 side by side for comparison. Merge the two figures perhaps?
Bar plots in Fig. 7 and 8 are hard to read. Having concrete values for each bar, and mentioning the specific values of $M$ and $T$ used would be very useful.

### Grammar/Language:

**Abstract**
- "The brain is targeted..." is not well formed.
- "We demonstrate our model..." -> "We demonstrate that our model..."

**Sec 1:**
- "as we experience the 'mental time travel'" -> "as we experience 'mental time travel'"
- "By a sequence attractor, it means"
- "The algorithm is proved to converge" -> "The algorithm is proven to converge"
- ...

There are many more. I would suggest passing the text through a grammar check.

---

> ### Author Response · Authors · 2023-11-18
>
> We thank the reviewer for the encouragement and helpful suggestions on improving our paper. Below, we address each point of yours.
>
>
> * **A discussion of the capacity of the proposed architecture is missing**
> In the revised paper, we derived the lower bound of the capacity of our network model. The maximal length of sequences that can be stored is at least $M$ (number of hidden neurons).  Please see the updated Proposition 1 in the revised paper.
>
>  * **The experiments section also does not have sufficient details about hyper-parameters**
> We have tested different hyper-parameters and found the results are consistent. Meanwhile, we provide the code in the supplementary material for reproduction of our experiments.
>
> * **How many times is each sequence presented to the model?**
>     In training, we ran our algorithm for maximal 500 epochs. Each epoch correspond to presenting the sequences to the model once. Empirically, we found the algorithm usually converges within 100 epochs.
>
> * **What's the capacity of the model? How many sequences can it store?**
>     The maximal length of an arbitrary sequence that can be stored in the model is at least $M$ (number of hidden neurons). Please see the updated Proposition 1 in the revised paper. The model can store as many sequences as possible as long as the total length of the sequences is within the model capacity.
>
> * **The sign function and the Heaviside function seem to be identical the way it's been defined in the paper.**
>     Both functions are needed. The sign function is to simplify the math such that one does not have to explicitly convert the values of network states from $\{0,1\}$ to $\{-1,1\}$, as in the classical Hopfield networks. The Heaviside function is needed as $\mu_i(t)$ and $\nu_j(t)$ are errors which need to be non-negative.
>
> * **The first sentence of Related work just lists a bunch of papers, which seems redundant, since these papers are explained later anyway.**
>     We modified the writing as you suggested in the revised paper.
>
> * **Would have been useful to have Fig. 2 and Fig. 5 side by side for comparison. Merge the two figures perhaps?**
>     We considered this suggestion but found merging two figures would break the logic line of our presentation for the first-time readers.
>
> * **Bar plots in Fig. 7 and 8 are hard to read. Having concrete values for each bar, and mentioning the specific values of and used would be very useful.**
>     We considered this suggestion but found adding specific values in the figure would make the figures visually messy.
>
>
> * **I would suggest passing the text through a grammar check.**
>     We thank the reviewer for the writing suggestions and incorporated them in the revised paper. We will polish our paper further by a large language model later.

---

> > ### Comment · Reviewer_9sz4 · 2023-11-20
> >
> > Thank you for your responses and clarifications. Proposition 1 further strengthens the paper.
> >
> > Reg. Fig. 7 and 8, I would encourage the authors to at least have a table in the supplement with all the values so that the readers are not left to guess the specific values based on the bar plots.

---

> > > ### Author Response · Authors · 2023-11-21
> > >
> > > Thank you so much for the suggestion. We have added two tables for Figure 7 and 8 in the PDF file of the updated supplementary material.

---

### Official Review · Reviewer_ZADc · 2023-11-06

**Soundness:** 2 fair
**Presentation:** 3 good
**Contribution:** 1 poor
**Rating:** 3
**Confidence:** 4

**Summary:**

This paper presents a new binary RNN in which there is a hidden layer in between RNN transitions. They develop a learning rule for training the two weight matrices in the transition. They then train these RNNs on some simple sequences and show they can be learned.

**Strengths:**

The paper is clearly presented, and the work is original to my knowledge. My main concern is its significance (see below).

**Weaknesses:**

You can always add more neurons to solve these problems with a standard RNN. See Siegelmann and Sontag (1992). You’d need to actually show that the other RNNs can’t actually learn these sequences, but it’s clear that they could…(maybe not a size matched RNN, but a larger one could…)

You’re training a new network for each sequence? This is pretty extreme, and there are lots of much simpler ways of solving that problem (e.g. a HMM).

In sum, it’s not really clear that the proposed method actually solves a real problem. You’d need to show it by comparing performance to other RNN architecture (as well as non sized matched networks). You’d also need to test against networks that don’t just have binary outputs. I understand that you would like to relate it to the brain, however the brain communicates in spike rates, not just a single spike…

**Questions:**

See weaknesses

---

> ### Author Response · Authors · 2023-11-11
> **The reviewer misunderstood the paper**
>
> We thank the reviewer for the feedback. However, the reviewer seems to misunderstand our paper in several aspects.
>
> 1. Fundamentally, we are not trying to solve a "real problem" in the sense of engineering applications, but to answer a biological question: how does the brain learn to store and retrieve sequence memory? Storing the sequences without biologically plausible mechanisms is not the interest of our paper.
>
> 2. Given a sequence $\mathbf{x}(1),...,\mathbf{x}(T)$, our goal is not simply storing it as it is, but storing it **as an attractor** such that given a perturbed state $\hat{\mathbf{x}}(t)$, the network can recover the rest of the stored sequence robustly. Please refer to Figure 9 and 10 and the supplementary material for such a demonstration.
>
> 3. While the standard RNN in deep learning literature can certainly store the sequences, it is trained with backpropagation, whose biological plausibility is in question. While in our work, the algorithm has a clear biological interpretation.
>
> 4. Our focus is on theoretical analysis. We proved how the algorithm can converge and give rise to sequence **attractors**. Such results are missing for the standard RNN trained with backpropagation.
>
> Below, we address several of your points.
>
> > You’re training a new network for each sequence?
>
> No. For the Moving MNIST experiment, we use a single network for storing 20 sequences.
>
> > the brain communicates in spike rates, not just a single spike…
>
> It is questionable that the brain communicates only in spike rates. There is a classic debate between rate coding and temporal coding. As to single spikes, please refer to the following paper for their role in neural coding:
>
> The Role of Spike Timing in the Coding of Stimulus Location in Rat Somatosensory Cortex (Panzeri et al., Neuron 2001)

---

> ### Author Response · Authors · 2023-11-18
> **Comparison experiments on networks of continuous neurons**
>
> We conducted experiments on networks of continuous neurons trained with backpropagation, in response to the reviewer's concern. The results are in Appendix A of the revised paper. Our algorithm converges to zero training error much faster empirically.

---

> > ### Comment · Reviewer_ZADc · 2023-11-23
> > **Thanks for your responses.**
> >
> > Thanks for your responses. I still don't understand how you manage to do multiple sequence when there may be the same sequence element across sequences? Or are sequence elements not repeated? This was the crux of my complaint, you are mapping x(t) -> x(t+1) so it's not possiible to build a latent representations that can deal with aliased sequence elements. Otherwise you're basiacally just mapping x(t) -> x(t+1) with a hidden layer inbetween. Perhaps I'm wrong, in which case I'd be happy to raise my score.

---

> ### Author Response · Authors · 2023-11-23
>
> Our current work cannot handle multiple sequences when there may be the same sequence element across sequences. The related work in comparison cannot handle such case either. For example,
>
> Learning patterns and pattern sequences by self-organizing nets of threshold
> elements. Amari. IEEE Transactions on Computers, 1972.
>
> Perceptron-like learning in time-summating neural networks. Bressloff and Taylor.
> Journal of Physics A: Mathematical and General, 1992.
>
> Long sequence hopfield memory. Chaudhry et al. NeurIPS, 2023.
>
> This can be handled by considering time delay which maps $\mathbf{x}(t), \mathbf{x}(t-1),...,\mathbf{x}(t-\tau)$ to $\mathbf{x}(t+1)$ or introducing stochastic neurons and is left to the future work.

---

> > ### Comment · Reviewer_ZADc · 2023-11-23
> >
> > OK now I get it. It's not about RNNs at all. It's a way of increasing the capacity of stored paired associations in a Hopfield-like networks, and just that the paired associations come in a sequence. You shouldn't really call it RNNs as RNNs were developed precisely for dealing with latent states (i.e. not confusing two identical sequence elements from two different sequences). I'm happy to incease my score provided the authors make it clear that it's not a RNN, i.e. by removing 'recurrent networks' from the title, and making clear in the main text that the setup is rapid binding of paired associations and not anyhting like a RNN.

---

> ### Author Response · Authors · 2023-11-23
>
> We use the term "recurrent" to refer to a neuron's output is feedback to itself in a biological sense. As it causes confusion, we would like to consider changing the title to, for example,
>
> Learning Sequence Attractors in Associative Networks with Hidden Neurons
>
> Thank you so much for the suggestion!

---

> > ### Comment · Reviewer_ZADc · 2023-11-23
> >
> > Yes that would be better - thanks!

---

> > > ### Author Response · Authors · 2023-11-23
> > >
> > > We have changed our title to **Learning Sequence Attractors in Hopfield Networks with Hidden Neurons** and revised the text accordingly. Please see our revised paper. Thank you again for the suggestion.

---

### Author Response · Authors · 2023-11-18
**To all reviewers and AC**

We are grateful to all the reviewers for their proofreading and suggestions on improving our paper. We incorporated the suggestions in the revised paper, which has two major additional features:

* **Network capacity bound**
We updated Proposition 1 in the revised paper. Now, it gives an explicit network capacity bound. The maximal length of an arbitrary sequence that can be stored in the network model is at least $M$ where $M$ is the number of hidden neurons.

* **Comparisons to other network models**
In Appendix A and B, we compared our network model to networks of continuous neurons and to modern Hopfield networks for learning the sequences in Section 6.3. We listed several advantages of our approach in comparison.

We are delighted to answer any further question and engage in the discussion.

---

### Meta-Review · Area_Chair_KUbM · 2023-12-11

**Metareview:**

This paper proposes a new approach for learning sequence attractors in recurrent networks with hidden neurons and a local learning algorithm, aiming to address how the brain stores and retrieves sequence memories. The authors demonstrate the robustness and efficacy of their model in learning and retrieving sequences, providing a theoretical foundation and experimental validation with both synthetic and real-world datasets.

The primary strengths of the paper include its originality, clarity of presentation, and potential implications for understanding neural information processing in the brain. However, concerns were raised about the novelty of the approach compared to existing hierarchical attractor network architectures, the lack of extensive comparisons with other methods, and biological implications of the model.

**Justification For Why Not Higher Score:**

Concerns raised about the novelty of the approach compared to existing hierarchical attractor network architectures, the lack of extensive comparisons with other methods, and biological implications of the model.

**Justification For Why Not Lower Score:**

N/A

---

### Decision · Program_Chairs · 2024-01-16

Reject